# Effect of Liquid Blood Concentrates on Cell Proliferation and Cell Cycle- and Apoptosis-Related Gene Expressions in Nonmelanoma Skin Cancer Cells: A Comparative In Vitro Study

**DOI:** 10.3390/ijms252312983

**Published:** 2024-12-03

**Authors:** Eva Dohle, Lianna Zhu, Robert Sader, Shahram Ghanaati

**Affiliations:** FORM, Frankfurt Orofacial Regenerative Medicine, Department for Oral, Cranio-Maxillofacial and Facial Plastic Surgery, Medical Center of the Johann Wolfgang Goethe University, 60590 Frankfurt, Germany; lianna97@foxmail.com (L.Z.); sader@em.uni-frankfurt.de (R.S.); s.ghanaati@med.uni-frankfurt.de (S.G.)

**Keywords:** nonmelanoma skin cancer, blood concentrates, cell proliferation, apoptosis, cell cycle

## Abstract

Nonmelanoma skin cancer (NMSC) presents a significant challenge to global healthcare due to its rising incidence, prompting the search for innovative treatments to overcome the limitations of current therapies. Our study aims to explore the potential effects of the liquid blood concentrate platelet-rich fibrin (PRF) on basal cell carcinoma cells (BCCs) and squamous cell carcinoma cells (SCCs) in order to obtain results that may lead to new possible adjunctive therapies for managing localized skin cancers, particularly NMSC. Basal cell carcinoma (BCC) cells and squamous cell carcinoma (SCC) cells were indirectly treated with PRF generated via different relative centrifugation forces, namely high and low RCF PRF, for 7 days. PRF-treated cells were comparatively analyzed for cell viability, proliferation and cell cycle- and apoptosis-related gene expression. Analysis of MTS assay results revealed a significant decrease in cell viability in both BCC and SCC cells following PRF treatment for 7 days. Ki-67 staining showed a decreased percentage of Ki-67-positive cells in both BCC and SCC cells after 2 days of treatment compared to the control group. The downregulation of CCND1 gene expression in both cell types at 2 days along with the upregulation of p21 and p53 gene expression in SCC cells demonstrated the effect of PRF in inhibiting cell proliferation and inducing cell cycle arrest, especially during the initial phases of treatment. Increased expression of caspase-8 and caspase-9 was observed, indicating the activation of both extrinsic and intrinsic apoptotic pathways by PRF treatment. Although the exact immunomodulatory properties of PRF require further investigation, the results of our basic in vitro studies are promising and might provide a basis for future investigations of PRF as an adjunctive therapy for managing localized skin cancers, particularly NMSC.

## 1. Introduction

Nonmelanoma skin cancer (NMSC) is the most common malignancy among Caucasian populations, with its global incidence steadily rising due to factors such as population aging and increased exposure to ultraviolet (UV) radiation resulting from lifestyle changes [1]. This trend imposes a significant economic burden on national healthcare systems and presents a substantial public health challenge worldwide [2]. Basal cell carcinoma (BCC) and squamous cell carcinoma (SCC) constitute the predominant subtypes of NMSC, collectively representing the vast majority of cases. The majority of NMSC is highly curable, particularly if diagnosed in the early stages. The pathogenesis of BCC and SCC is complex and multifactorial. Both types of skin cancer are characterized by a high burden of mutations induced by ultraviolet radiation. UV-induced DNA damage leads to mutations in key genes involved in cell cycle regulation, cell proliferation, DNA repair, and apoptosis. Accumulation of these genetic alterations and dysregulation of signaling pathways ultimately initiate the development of NMSC [1,3]. BCC and SCC cells exhibit different genetic alterations driving their tumorigenesis. BCC cells commonly arise from mutations that activate the Hedgehog (Hh) signaling pathway [4,5] whereas SCC cells are characterized by mutations in genes such as TP53 (Tumor protein P53), NOTCH1 (Neurogenic locus notch homolog protein 1), and CDKN2A (cyclin-dependent kinase inhibitor 2A) [6].

The most effective way of preventing the accumulation of mutations in the skin and the development of skin cancers is to minimize exposure to UV radiation [7]. Although current treatments for NMSC (including surgery, radiotherapy, and topical medications) are effective in many cases, they have limitations such as invasiveness and side effects, including scarring, pain, and local skin reactions. Therefore, novel treatments are required to address these challenges [8,9,10,11,12]. Platelet-rich fibrin (PRF), as the second-generation blood concentrate, is an autologous biomaterial widely used in various medical fields such as dentistry, dermatology, and orthopedics [13,14,15,16]. Numerous studies highlight the extensive dermatological applications of blood concentrates including treating chronic wounds, skin rejuvenation, vitiligo, and hair restoration [17,18]. It is increasingly being adopted across various new areas of dermatology, either as a standalone therapy or in combination with traditional treatments. The bioactivity of PRF seems to be driven by its autogenous nature and the concentrated delivery of immune cells including platelets, leukocytes, and various plasma proteins embedded in a fibrin matrix, serving as a reservoir for growth factors and cytokines and enhancing immune responses. The understanding that tumor cells are cells that can evade immunity through mutations leads to the idea that a strengthened immune system combined with other treatments might help to eliminate the abnormal cells or prevent them from growing further. Platelets, the primary component of PRF, are capable of releasing a series of growth factors, including PDGF, TGF-β1, VEGF, and EGF, and cytokines such as interleukin (IL)-1β, IL-6, IL-4, and TNF-α, which play crucial roles in angiogenesis, tissue regeneration, and wound healing [19]. PRF has been shown to have effects on human fibroblast and osteoblast proliferation and activation, as well as on vessel-like structure formation when combined with endothelial cells [16,20]. However, despite these benefits, its effects on cancer cells remain underexplored. This basic study sought to investigate the impact of PRF on NMSC cells in vitro, building upon previous research from our group that demonstrated reduced cell proliferation and increased cell death in osteoblastic and fibroblastic tumor cell lines treated with PRF, which suggests that PRF may have potential as an adjunctive therapy for localized tumors [21]. In this context, our study aims to investigate the potential effects of PRF on SCC and BCC cells in order to obtain results that may enhance the exploration of new possible adjunctive therapies for the treatment of localized skin cancers, particularly NMSC, taking advantage of its benefits in wound healing and tissue regeneration as well as its autologous nature. In this study, we performed in vitro experiments treating BCC and SCC cells indirectly with PRF over a 7-day period to investigate its potential tumor-suppressive effects in NMSC cells.

## 2. Results

### 2.1. PRF-Mediated Decrease in BCC and SCC Cell Viability

To analyze the effect of PRF treatment on the cell viability of BCC and SCC, an MTS assay was performed after treatment with high and low RCF PRF or without PRF for 2, 4, and 7 days. The results (Figure 1) represent relative cell viability calculated by comparing PRF-treated cells with untreated cells in the control group. After two days of PRF treatment, a statistically significant reduction in cell viability was observed in both BCC and SCC, although the reduction was more pronounced in BCC than in SCC (A). After four days, PRF-treated BCC still showed a statistically significant decrease in cell viability, while the cell viability of PRF-treated SCC was also reduced, but not statistically significant (B). By day 7, cell viability was also reduced in both cell types, with a statistically significant reduction in SCC cells. The effect on both cell types was relatively similar (C). In addition, no apparent difference was observed between cells treated with high or low RCF PRF at any of the three time points for either cell type (A, B, C). Notably, the effect of PRF on cell viability was most pronounced on day 2 in BCC, with approximately 50% relative cell viability compared to the control group. Over time, the effect of PRF appeared to gradually decrease, reaching approximately 80% relative cell viability by day 7 (D). Similarly, in SCC, the most pronounced effect of PRF was observed on day 2. However, the overall measured values in SCC remained relatively stable, consistently around 80% (E).

### 2.2. PRF Treatment of BCC and SCC Results in Decrease in Ki67-Positive Cells

To examine the effect of high and low RCF PRF treatment on the cell proliferation of BCC and SCC cells, the cells were immunofluorescently stained using Ki67 after 2 days of high and low RCF PRF treatment, as well as without PRF treatment (control). Figure 2 shows that the control group exhibited more Ki67-positive cells than the high and low RCF PRF-treated groups in both cell types, with a more pronounced effect in SCC. There was no significant visual difference observed between cells under high and low RCF PRF treatment in both BCC and SCC cells (Figure 2C,D). The data presented in Figure 2C,D show that the percentage of Ki67-positive cells in the PRF-treated group was significantly lower than in the control group. QuPath software (v. 0.5.1) was used for quantification of Ki67-positive cells and the resulting data are presented as relative percentages. The effect of high and low RCF PRF on BCC cells seems to be similar, whereas in SCC cells, the effect of high RCF PRF appears to be slightly but not significantly greater than that of low RCF PRF, resulting in a lower percentage of Ki67-positive cells when treated with high RCF PRF.

### 2.3. Effect of High and Low RCF PRF Treatment on Cell Cycle- and Apoptosis-Related Gene Expressions in BCC and SCC

To investigate the effect of PRF treatment on cell cycle and apoptosis in BCC and SCC cells, gene expressions of related genes were analyzed after treatment with high and low RCF PRF for 2, 4, and 7 days and compared to the gene expression of these genes in untreated cells in the control group. The analyzed genes included caspase 8 and 9, cyclin D1 (CCND1), p21, p53, and RPL37A as the endogenous control. The results are presented as relative gene expression compared to the control.

As a key mediator in the initiation of apoptosis through the extrinsic pathway, the effect of PRF treatment on the gene expression of caspase 8 (Figure 3) was analyzed. Although the PRF-mediated differences in the gene expression of caspase 8 are generally not significant due to high standard deviations, a clear trend in caspase 8 gene expression in response to PRF treatment can be detected. After two days of PRF treatment, the gene expression of caspase 8 was found to be upregulated in both BCC and SCC cells compared to the control group. The gene expression was higher under low RCF PRF treatment compared to high RCF PRF (A,D). The expression of caspase 8 gene remained upregulated in both cell types after four days of PRF treatment. In BCC, both high and low PRF treatments resulted in similar effects, whereas in SCC cells, a higher upregulation was observed under high PRF treatment (B,E). After 7 days, the gene expression of caspase 8 was slightly upregulated in BCC cells, while in SCC cells, particularly in response to low RCF PRF treatment, it was highly upregulated (C,F).

Caspase 9 plays an important role as an initiator caspase in the intrinsic apoptosis pathway. The effect of PRF treatment on the gene expression of caspase 9 (Figure 4) was analyzed. In BCC, caspase 9 gene expression was upregulated on day 2, with a statistically significant increase only when cells were treated with high RCF PRF treatment, but gene expression was decreased statistically significantly on days 4 and 7 after PRF treatment (A–C). However, in SCC, the expression of caspase 9 was consistently upregulated compared to the control group on days 2, 4, and 7 in response to PRF treatment. Statistically significant upregulation was observed in SCC at all time points examined, except for the fourth day when treated with high PRF treatment (not significant), with the effect of low PRF treatment being notably higher than that of high PRF treatment on days 4 and 7 (D–F).

CCND1 encodes cyclin D1. It forms cyclin–CDK complexes with CDKs and plays a crucial role in regulating the cell cycle, particularly promoting progression from the G1 phase to the S phase. The effect of PRF treatment on the gene expression of CCND1 (Figure 5) was analyzed. In BCC cells, CCND1 gene expression was generally downregulated at 2 d and 4 d under PRF treatment. Statistically significant differences were observed after high RCF PRF treatment at 2 d and low RCF PRF treatment at 4 d. All other changes in gene expression could not be calculated as statistically significant different. After 7 days, the gene expression was higher but not significantly different in the groups treated with high and low RCF PRF compared to the control group (A–C). In SCC, a statistically significant reduction in CCND1 gene expression was observed only after 2 days of low RCF PRF treatment and after 7 days of high RCF PRF treatment (D,F).

p21, a crucial cell cycle regulator, arrests the cell cycle at the G1 and G2 phases by binding and inhibiting the complexes of CDK and cyclin. The effect of PRF treatment on the gene expression of p21 (Figure 6) was analyzed. In BCC, downregulation of p21 gene expression was observed at all three time points in response to PRF treatment. Although after 2 days of treatment, there was no significant difference in gene expression, a slight trend of downregulated gene expression could be observed (A). A statistically significant decrease in gene expression could be observed under low RCF PRF treatment at day 4 and under both high and low RCF PRF treatments at day 7. The effect of high and low RCF PRF seems to be similar at three-time points in BCC (A–C). Conversely, no significant differences in the gene expression of p21 could be evaluated after 2 days of SCC treatment, and upregulation of p21 gene expression was observed at day 4 and day 7 in SCC in response to both RCF PRF treatments. Statistically significant upregulation was noticed after low RCF PRF treatment on day 4 and under high and low RCF PRF treatment on day 7. The upregulation was notably higher in response to low RCF PRF treatment compared to high RCF PRF treatment (D–F).

p53 is a tumor suppressor gene, which prevents the uncontrolled cell growth and division of cells. It is a key regulator of the cell cycle, promoting cell cycle arrest at the G1 checkpoint. The effect of PRF treatment on p53 gene expression was analyzed (Figure 7). The gene expression of p53 in BCC was downregulated in response to PRF treatment at all three time points, but only statistically significantly decreased on the fourth day, with lower levels of gene expression in the low RCF PRF-treated group (A–C). In contrast, in SCC, gene expression of the p53 gene was similar to the control group on the second day, followed by upregulation on the fourth and seventh days. On the fourth day, the upregulation of low RCF PRF was higher than that of high RCF PRF, while on the seventh day, the upregulation of high RCF PRF was again higher than that of low RCF PRF (D–F). None of the differences in gene expression in SCC treated with PRF could be assessed as significantly different.

## 3. Discussion

The increasing prevalence of NMSC and its substantial impact on healthcare systems worldwide highlight the necessity for novel treatment approaches [22,23]. Current treatment modalities for NMSC, such as surgery, radiation therapy, chemotherapy, and topical pharmacological treatments, are often associated with limitations such as invasiveness and side effects like scarring, pain, and local skin reactions ranging from mild irritation to severe inflammation [5,8,10,11,12,24]. The use of PRF has shown significant promise across various medical fields, including dentistry, oral surgery, dermatology, pain management, and plastic surgery. PRF’s bioactivity enhances immune responses and supports tissue regeneration and wound healing [14,15,16,20,25,26,27,28,29]. Our study aims to explore the potential tumor-suppressive effects of PRF as a possible adjunctive therapy for localized skin tumors, particularly NMSC, by enhancing immune responses and improving patient outcomes. Specifically, we investigate the impact of PRF treatment on BCC and SCC cells in vitro, focusing on cell viability, proliferation, cell cycle progression, and apoptosis-related factors following indirect treatment with injectable PRF generated at high and low RCF levels over a 7-day period.

Notably, although not calculated as significant for all time points, PRF treatment resulted in a general trend of a decrease in cell viability in both cell types compared to untreated cells. These findings suggest the potential suppressive effect of PRF treatment on cellular activity and metabolic processes in BCC and SCC cells. Platelets and leukocytes serve as the main cells responsible for the biological activity of PRF [30]. Leukocytes, including neutrophils and monocytes, play crucial roles in the inflammatory response by contributing to the removal of pathogens and modulating immune reactions. Neutrophils, acting as the first line of defense against infections, exhibit selective cytotoxicity against tumor cells in vitro [31], potentially through the release of reactive oxygen species (ROS) and proteolytic enzymes. Recent studies have highlighted the continuous release of growth factors and cytokines from liquid PRF, such as PDGF, TGF-β1, VEGF, and EGF, which are crucial in various cascades of angiogenesis, regeneration, and wound healing [27,28,30,32]. Platelet–neutrophil interactions enhance the oxidative burst, promote neutrophil extracellular trap formation, and enhance phagocytosis, which is critical for host defense mechanisms [33,34]. The cytotoxic effects of human platelets on tumor cells have been already observed in vitro [35,36]. Dysregulated cell proliferation is a hallmark feature of cancer and is driven by a complex interplay of genetic, molecular, and environmental factors. To examine the effect of PRF treatment on the cell proliferation of BCC and SCC cells, immunofluorescent staining using Ki-67 was performed after 2 days of high and low RCF PRF treatment, as well as in the absence of PRF treatment (control). In the resulting immunofluorescence images, the control group exhibited visibly more Ki-67-positive cells compared to both high and low RCF PRF-treated groups in both BCC and SCC cell types, with a particularly pronounced effect noted in SCC cells. No significant visual difference was observed between cells under high and low RCF PRF treatment in BCC and SCC. These observations were further confirmed by quantification analysis, which revealed a lower percentage of Ki-67-positive cells in the PRF-treated groups compared to the control group in both cell types. Additionally, there was no significant difference in the percentage of Ki67-positive cells between cells treated with low and high RCF PRF in both cell types, indicating that the effect of PRF treatment on cell proliferation in both BCC and SCC cells is independent of the level of RCF during PRF preparation. Ki-67 is a protein expressed during active phases of the cell cycle, including G1, S, G2 phase, and mitosis, but not in resting cells (G0 phase) [37]. Its expression levels correlate with the rate of cell proliferation, making it a valuable marker for assessing the proliferative activity of tumor cells. Interestingly, the expression of Ki67 is increased in sun-damaged skin [38] and a significantly high incidence of Ki67-positive cells in malignant epithelial tumors of the skin has been reported [39], indicating its potential as a reliable marker for BCC and SCC. Our findings suggest that PRF treatment potentially induces a shift in the cell cycle dynamics of both BCC and SCC cells, leading to reduced proliferative activity. This shift towards a higher proportion of cells in the resting phase (G0 phase) of the cell cycle indicates a slowdown or inhibition of cell proliferation [40].

To investigate the potential mechanism of PRF in reducing the viability of BCC and SCC, we analyzed the expression of apoptosis-related genes, such as caspase 8 and caspase 9, in PRF-treated cells compared to untreated cells. In summary, our findings indicate that PRF treatment may potentially exert a tumor-suppressive effect by upregulating caspase 8 and caspase 9 gene expression, leading to the activation of both intrinsic and extrinsic apoptosis pathways. One possible explanation for this effect could be the increase in proapoptotic stimuli such as cytotoxic T lymphocytes, granzymes B, Fas, TNF, and Bax, among others [41,42,43,44]. This effect of activating both intrinsic and extrinsic apoptosis pathways appears to persist in SCC cells throughout the experimental period. However, in BCC cells, while the extrinsic apoptosis pathway may remain activated for 7 days, the activation of the intrinsic pathway may be limited to 2 days. This indicates a potential evasion of intrinsic apoptosis or cellular adaptation to PRF treatment in BCC cells over time. Additionally, there was no notable difference observed between high and low RCF PRF treatment in apoptosis-related gene expressions in both cell types, suggesting that this tumor-suppressive effect of PRF-induced apoptosis is not significantly influenced by the level of RCF generated during PRF preparation. However, it is essential to note that these findings are based on in vitro studies and may not fully reflect the complexity of the in vivo tumor microenvironment. In particular, the early onset of NMSC may have considerable genetic consequences on different genes and molecular mechanisms [45]. These alterations in the functioning of essential signaling pathways in cell growth control, like the Sonic Hedgehog pathway, continue to drive tumor progression [46]. Although these in vitro results are very promising, this study has to be considered as initial basic research, and further research is needed to investigate the mechanism of PRF’s effect on apoptosis in tumor cells. The similar patterns of expression of p53 and p21 in response to PRF treatment in each cell type indicate a potential interplay between these two molecules in the cellular response to PRF. The upregulation of both p53 and p21 in SCC cells following PRF treatment suggests an active effect of PRF on the p53/p21 signaling pathway, potentially enhancing the tumor suppressive function of these genes, inducing apoptosis, and promoting cell cycle arrest. Conversely, the downregulation of both p53 and p21 in BCC after 4 days of PRF treatment suggests a negative impact of PRF on the p53/p21 signaling pathway, which could imply a shift in the cellular response towards evading apoptosis and escaping cell cycle arrest. This phenomenon may indicate a potential resistance of BCC cells to the antiproliferative effects of PRF treatment over time. Although the results are very promising from a clinical point of view, NMSC patients often exhibit elevated levels of reactive oxygen species (ROS) and systemic inflammation in their bloodstream [47,48]. This might impair platelet function, potentially reducing the efficacy of PRF, altering PRF’s structural integrity and the release kinetics of growth factors. Further investigation is needed, carefully considering potential limitations, to better understand the mechanisms underlying the potential effect of PRF on BCC and SCC cells.

## 4. Materials and Methods

### 4.1. Ethical Statement

The generation and application of the autologous blood concentrates that were used for this study were in accordance with the principle of informed consent and approved by the responsible Ethics Commission of the state Hessen, Germany (265/17). All donors gave written informed consent to use their blood for study purposes and were only included when healthy, aged ≥ 18 and without any relation to the principal investigator. Exclusion criteria were corticosteroid intake, ongoing chemotherapy or radiotherapy, infectious disease, anticoagulation therapy, and pregnancy.

### 4.2. Cell Lines

The human BCC cell line (BCC-1/KMC) was kindly provided from Dr. Chia-Yu Chu, National Taiwan University [49] (Taipei, Taiwan). BCC cells were cultured in T-175 cell culture flasks in RPMI 1640 medium, supplemented with 10% FBS and 1% Pen/Strep. The human SCC cell line (SCC-25) was obtained from the Leibniz Institute DSMZ—German Collection of Microorganisms and Cell Cultures GmbH (Cat.no ACC 617). SCC cells were grown in T-175 cell culture flasks containing DMEM/F-12 supplemented with 10% FBS and 1% Pen/Strep. All cells were cultivated at 37 °C in a humidified atmosphere with 5% CO_2_. After reaching confluence, cells were detached using 0.25% Trypsin–EDTA and counted using Trypan blue and Hemocytometer for use in the study. For each experiment, 3 different cell cultures per cell line were used (independent replicates).

### 4.3. Preparation of PRF

Peripheral blood was collected from the antecubital vein of three healthy donors (male and female) aged between 20 and 50 years (median 34 years). All donors provided informed consent for the use of their blood in this study. Blood samples were collected in 10 mL sterile, plain, plastic PRF tubes and centrifuged immediately at 600 rpm (44 g) and 8 min for low RCF PRF and 2400 rpm (710 g) and 8 min for high RCF PRF preparation. After centrifugation, the resulting liquid low and high RCF PRF were collected and homogenized separately for each donor in sterile 15 mL plastic tubes and were used for the experiments, as described in the following section.

### 4.4. PRF Treatment of BCC and SCC

BCC and SCC cells were pre-seeded on Thermanox coverslips (Thermo Scientific, Karlsruhe, Germany) in 24-well plates (1 × 10^5^ cells/well) with 1 mL of medium (RPMI medium for BCC cells and DMEM/F-12 medium for SCC cells). After 24 h incubation, the medium was changed, and transwell inserts with a pore size of 0.4 μm (Greiner Bio-one, Frickenhausen, Germany) were placed over the seeded cells. Fresh low and high RCF PRF were prepared as described above. Then, 100 μL of low or high RCF PRF was pipetted into the transwell inserts, and after clotting, 100 μL of medium was added. Cells without the addition of PRF served as the control group. Plates were incubated in a CO_2_ incubator at 37 °C for 2 days, 4 days, and 7 days, with the medium changed every 2 days. For PRF treatment experiments, 3 different donors of PRF were used for each experimental group and each time point, and 3 independent experiments per experimental group and PRF donor were performed (in total n = 9). At each designated time point, transwell inserts were removed and cells were fixed for immunofluorescence staining. During this experimental setup, appropriate cells were treated with PRF only once for the whole cultivation time.

### 4.5. Cell Viability Assay (MTS)

Cell viability of SCC and BCC was determined by the MTS assay (CellTiter 96^®^ Aqueous One Solution Cell Proliferation Assay, Promega, Walldorf, Germany) following the manufacturer’s instructions. After removal of the transwell inserts, the wells were washed with PBS and replenished with 500 μL of fresh medium. Then, 100 μL of CellTiter 96^®^ AQueous One Solution Reagent was added to each well. Plates were incubated in a CO_2_ incubator at 37 °C for 1 h. After incubation, 100 μL from each well was transferred to a new 96-well plate for measurement. Absorbance was measured at a wavelength of 490 nm using a microplate reader (Infinite M200, Tecan, Zürich, Switzerland).

### 4.6. Immunofluorescence Staining and Quantification of Ki67 Positive Cells

Since Ki67 is a widely used marker for assessing cell proliferation, as it is expressed in all active phases of the cell cycle but not in resting cells, PRF-treated cells were stained for Ki67 and compared to control cells. Cells on coverslips were fixed with 4% formaldehyde, washed three times with PBS, and then permeabilized with 0.1% Triton-X100 in PBS for 5 min at room temperature to improve access to intracellular antigens. After rinsing with PBS, the samples were incubated with the primary antibody Ki67 (1:200 dilution; Thermo Fisher Scientific, Karlsruhe, Germany, Cat #MA5-41135) diluted in 1% BSA/PBS for 1 h at room temperature. After washing off the primary antibody, the cells were incubated with the fluorescently labeled secondary antibody (Alexa Fluor 546 anti-rabbit) diluted 1:1000 in 1% BSA in PBS for 1 h at room temperature, avoiding exposure to light. DAPI (diluted 1:1000 in 1% BSA in PBS) was then applied for nuclear counterstaining for 10 min at room temperature. After a final PBS wash, coverslips were transferred onto microscope slides and mounted with fluoromount aqueous mounting medium and microscope coverslips. Images were captured using the Eclipse Ni/E fluorescence microscope (556 nm; Eclipse TS 100, Nikon, Düsseldorf) with a DS-Ri2 camera (Nikon, Düsseldorf, Germany). For analysis, three fluorescence images were taken at 20× magnification from standardized locations on each slide within each experimental group, including the center, top left, and bottom right of the slide. QuPath software (v. 0.5.1) was used for image analysis. After defining the region of interest with the annotation tool bar, the cell detection command in QuPath was used to identify the number of cells in all nuclei automatically, based on nuclear staining in the blue DAPI channel. Cells were classified as positive if the intensity threshold of the detected cells was above 100 in the red Ki67 channel of the image, and positive cells were marked and colored differently than negative cells. With the detection measurement tool, the number of positive and negative cells was estimated, exported and calculated as the percentage of positive cells by dividing the number of positive cells by the total number of detected cells and multiplying by 100. The percentage of positive cells calculated from the PRF-treated group was compared to that of the control group for analysis.

### 4.7. Gene Expression Analyses

RNA isolation was performed using the RNeasy Micro Kit (Qiagen, Hilden, Germany) according to the manufacturer’s protocol and the concentration of RNA was measured with a NanoDrop spectrophotometer (Thermo Fisher Scientific, Karlsruhe, Germany). The isolated RNA was reverse transcribed into complementary DNA (cDNA) according to the standard protocol of Qiagen’s Omniscript^®^ reverse transcription kit (Qiagen, Hilden, Germany). Quantitative real time PCR, enabling the quantification of relative gene expression, was performed using SYBR green DNA binding fluorescent dye. The 16 μL SYBR green master mix for one reaction consisted of 2 μL Qiagen mix primer, 4 μL RNase-free water, and 10 μL SYBR green mix (Sigma, St. Louis, USA). It was mixed with 4 μL cDNA (1 ng/μL) in a total volume of 20 μL/well in a 96-well reaction plate. Quantitative real-time PCR was performed in triplicate. After initial activation of DNA polymerase at 94 °C for 2 min, 40 cycles were performed: 1. denaturation at 94 °C for 15 s, and 2. annealing and extension at 60 °C for 1 min. The primers that were used in this study include caspase 8 and 9, cyclin D1 (CCND1), p21, p53, and RPL37A (Table 1). The RPL37A gene was used as the endogenous control and the relative gene expression was determined using the comparative Ct method (∆∆Ct). The results are presented as fold change relative to RPL37A gene expression, and relative gene expression was compared by setting controls to 1 as a reference value (RQ control = 1).

### 4.8. Statistical Analysis

For each experimental group, samples from three independent PRF donors were analyzed, with three independent experiments conducted per donor (n = 9). The results were calculated as mean ± standard deviation (SD) and evaluated for significant differences with one-way analysis of variance (ANOVA) using MS Excel (Microsoft Office, Microsoft) and GraphPad Prism 9.0 software; differences were considered statistically significant for * *p*-value < 0.05, ** *p*-value < 0.01, *** *p*-value < 0.001 and **** *p*-value < 0.0001.

## 5. Conclusions

In conclusion, our findings suggest the antiproliferative effect of PRF in BCC and SCC cells by inducing cell cycle arrest at the early treatment stages. This is evidenced in this study by the downregulation of the negative cell cycle regulator CCND1 in both BCC and SCC at the early phase and the upregulation of the tumor suppressor genes p53 and p21 in SCC cells. However, potential adaptation or compensatory mechanisms may mitigate the downregulation of p53 and p21 expression in BCC cells and the upregulation of CCND1 in SCC cells over time. As no notable difference was exhibited in cell cycle-related gene expressions between high and low RCF PRF treatment in both cell types, it indicates that this tumor-suppressive effect of PRF in terms of inducing cell cycle arrest in BCC and SCC is not significantly influenced by the level of RCF generated during PRF preparation.

## Figures and Tables

**Figure 1 ijms-25-12983-f001:**
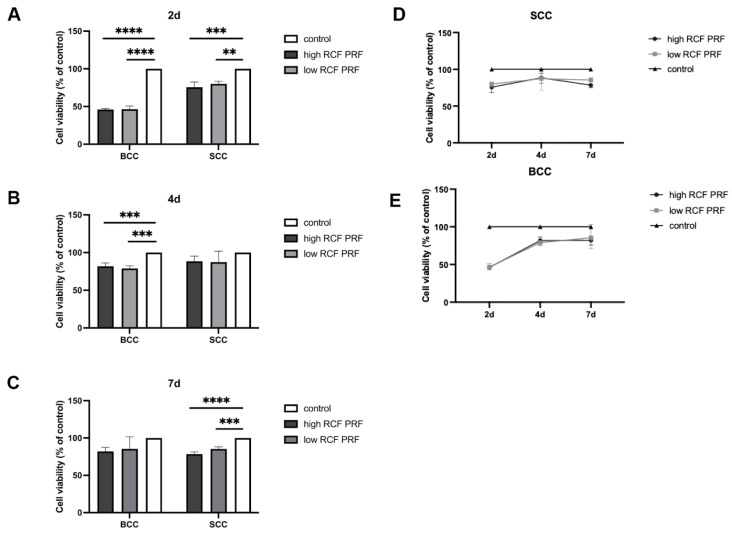
Cell viability of PRF-treated BCC and SCC compared to untreated cells. The cell viability of BCC and SCC treated with high and low RCF PRF or without PRF treatment as a control was assessed by the MTS assay at 2 days (**A**), 4 days (**B**), and 7 days (**C**). The line graphs (**D**,**E**) show the trend of the PRF effect on BCC and SCC cells over time. The relative cell viability percentage was calculated by comparing the cell viability of PRF-treated cells with that of untreated cells in the control group (=100%; n = 9). The bars represent the mean values and the corresponding standard deviations (SDs). Significance: ** *p* < 0.01, *** *p* < 0.001, and **** *p* < 0.0001.

**Figure 2 ijms-25-12983-f002:**
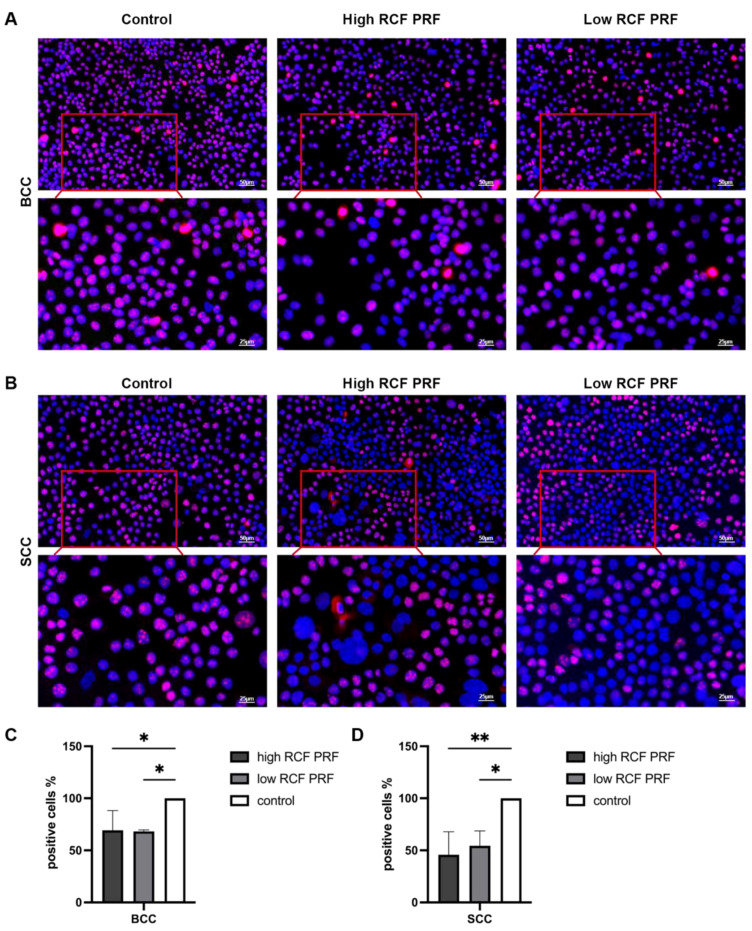
Analysis of immunofluorescence staining images of Ki67. Representative merged images of immunofluorescence staining of Ki67 from BCC (**A**) and from SCC (**B**) on day 2, treated with high and low RCF PRF, and without PRF treatment (control). Scale bars: upper row = 50 μm; lower row = 25 μm. The relative percentage of Ki67-positive cells in BCC (**C**) and SCC (**D**) was calculated by comparing the percentage of Ki67-positive cells of PRF-treated cells to that of untreated cells in the control group (n = 9). The bars represent the mean values and the corresponding standard deviations (SDs). Significance: * *p* < 0.05, ** *p* < 0.01.

**Figure 3 ijms-25-12983-f003:**
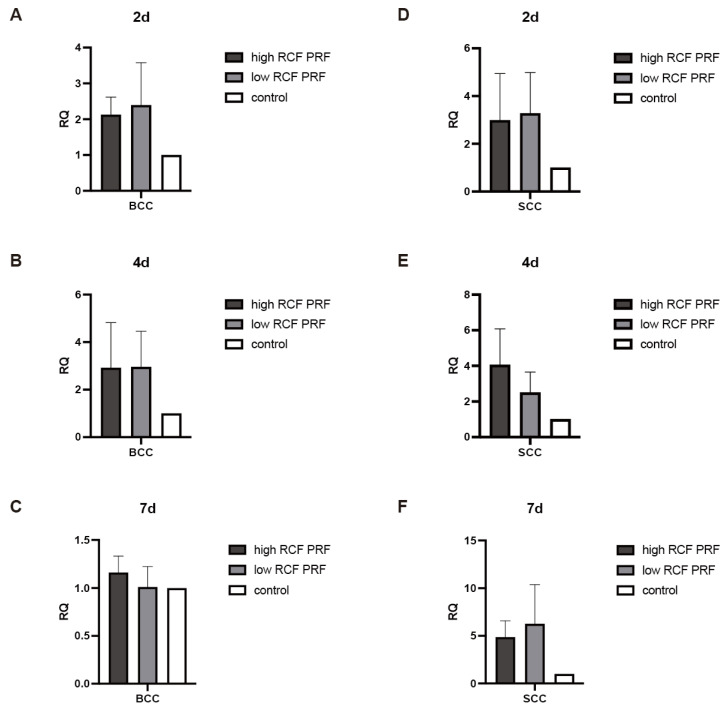
Relative gene expression of caspase 8. The relative gene expression of caspase 8 was assessed using quantitative real-time PCR. The expression levels of the caspase 8 gene in BCC (**A**–**C**) and SCC (**D**–**F**) in response to PRF treatment for 2, 4, and 7 days were compared with untreated cells in the control group (n = 9). None of the differences could be evaluated as statistically significant different. The results are presented as fold changes in gene expression relative to RPL37A.

**Figure 4 ijms-25-12983-f004:**
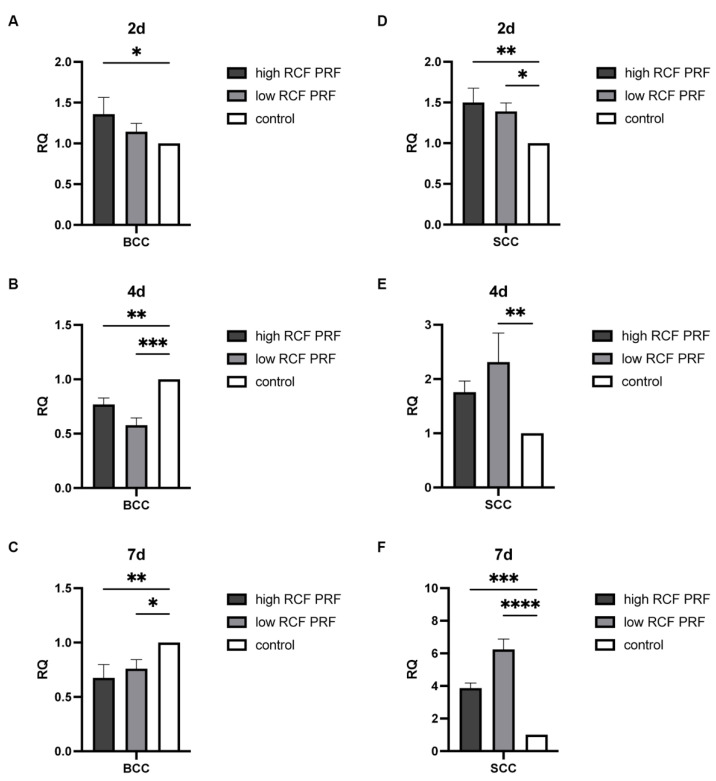
Relative gene expression of caspase 9. The relative gene expression of caspase 9 was assessed using quantitative real-time PCR. Caspase 9 gene expression levels in BCC (**A**–**C**) and SCC (**D**–**F**) treated with PRF for 2, 4, and 7 days were compared with untreated control cells (n = 9). The results are presented as fold changes in gene expression relative to RPL37A. Significance: * *p* < 0.05, ** *p* < 0.01, *** *p* < 0.001, and **** *p* < 0.0001.

**Figure 5 ijms-25-12983-f005:**
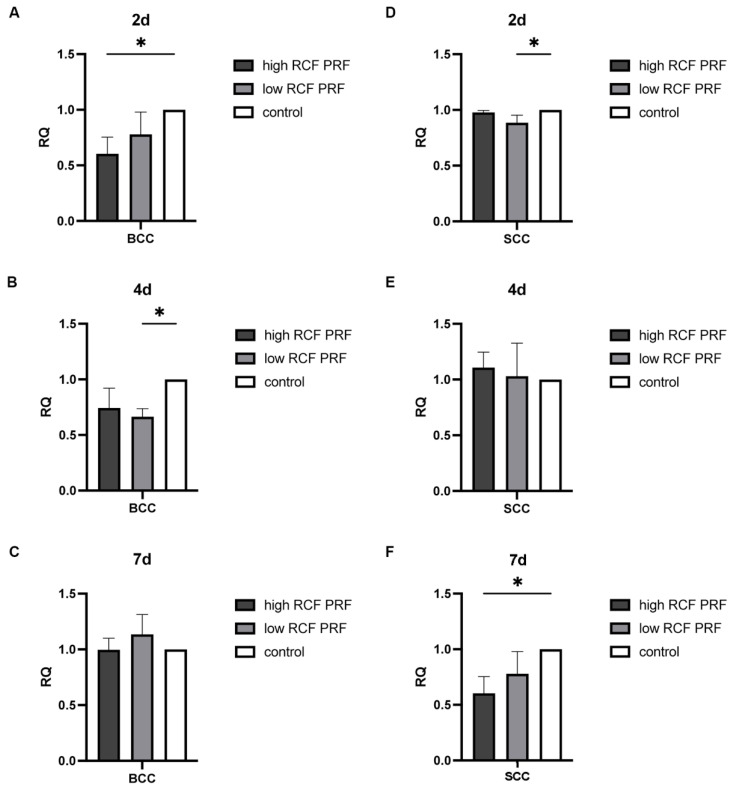
Relative gene expression of CCND1. The relative gene expression of CCND1 was assessed by quantitative real-time PCR. CCND1 gene expression levels in BCC (**A**–**C**) and SCC (**D**–**F**) treated with PRF for 2, 4, and 7 days were compared with untreated control cells (n = 9). The results are presented as fold changes in gene expression relative to RPL37A. Significance: * *p* < 0.05.

**Figure 6 ijms-25-12983-f006:**
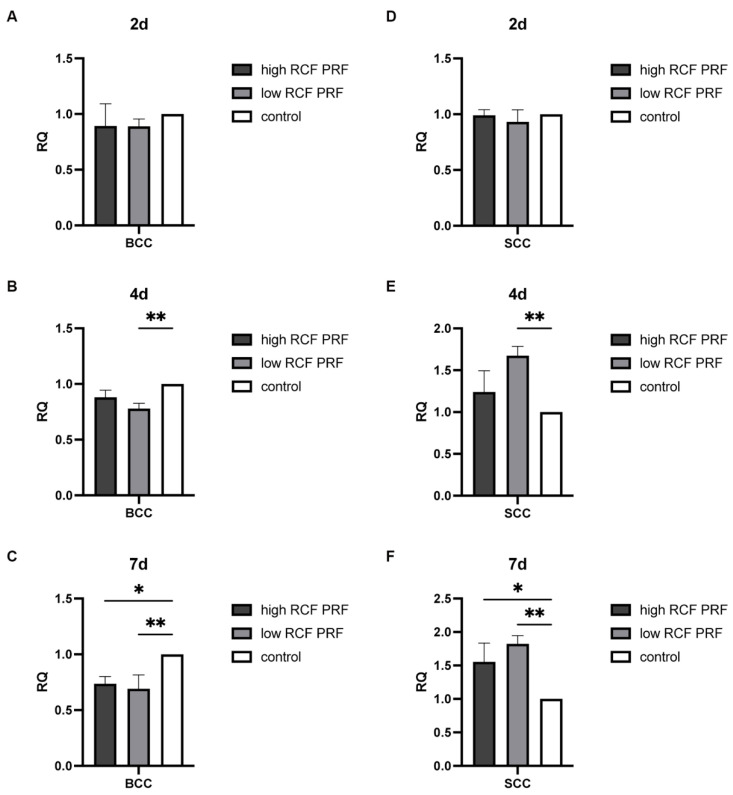
Relative gene expression of p21. Relative gene expression was assessed by quantitative real-time PCR. p21 gene expression levels in BCC (**A**–**C**) and SCC (**D**–**F**) treated with PRF for 2, 4, and 7 days were compared with untreated control cells (n = 9). The results are presented as fold changes in gene expression relative to RPL37A. Significance: * *p* < 0.05, ** *p* < 0.01.

**Figure 7 ijms-25-12983-f007:**
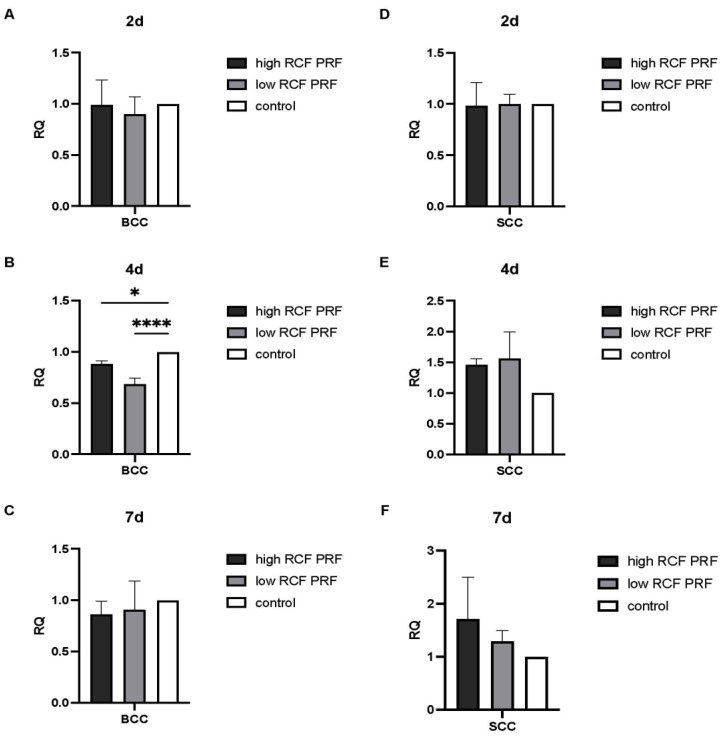
Relative gene expression of p53. Relative gene expression was assessed by quantitative real-time PCR. P53 gene expression levels in BCC (**A**–**C**) and SCC (**D**–**F**) treated with PRF for 2, 4, and 7 days were compared with untreated control cells (n = 9). The results are presented as fold changes in gene expression relative to RPL37A. * *p* < 0.05, **** *p* < 0.0001.

**Table 1 ijms-25-12983-t001:** Primer, primer sequences and cat number of primer assays used to evaluate gene expression.

Primer	Primer Assay Name	Sequence/Cat. Number
RPL13A		5′-TGT GGT TCC TGC ATG AAG ACA-3′ 5′-GTG ACA GCG GAA GTG GTA TTG TAC-3′
Caspase 8	Hs_CARD8_va.1_SG_QuantiTectPrimerAssay	Qiagen/QT02407188
Caspase 9	Hs_CASP9_1_SG_QuantiTectPrimerAssay	Qiagen/QT00036267
Cyclin D1	Hs_CCND1_1_SG_QuantiTectPrimerAssay	Qiagen/QT00495285
p21		5′-AAT GCG CAG GAA TAA GGA AG-3′ 5′-CGA GCT GTT TAC GTT TGA CG-3′
p53	Hs_CDIP1_1_SG_QuantiTectPrimerAssay	Qiagen/QT00003094

## Data Availability

The data that support the findings of this study are available on request from the corresponding author (E.D.).

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
