# Peer review of "Effect of Liquid Blood Concentrates on Cell Proliferation and Cell Cycle- and Apoptosis-Related Gene Expressions in Nonmelanoma Skin Cancer Cells: A Comparative In Vitro Study"

_ijms, 2024, doi:10.3390/ijms252312983_

Round 1
Reviewer 1 Report (New Reviewer)
Comments and Suggestions for Authors
I went thorough the manuscript titled “Immunomodulating effect of liquid blood concentrates on cell proliferation and cell cycle- and apoptosis-related gene expressions in nonmelanoma skin cancer cells: A comparative in vitro study”. Although this experimental work relies on a highly innovative idea of using Platelet-rich fibrin on SCC and BCC as an anti-malignant factor, the manuscript presents several flaws and ambiguities that limit my ability to assess the quality of the work and the validity of the results.
Below, I describe my suggestions in detail.
Line 2 The Word “Immunomodulating” should be excluded. In this study the researchers used cell lines to see expressions and viability. Cell cultures lack immune system and are not a suitable model for immunomodulating effects of any kind. Consider changing the title to describe your work. Since you have used Platelet-rich fibrin on SCC and BCC cell lines and measured gene expressions and cell viability that is you should report to the title
Lines 15 – 17 “Our study aims to explore the potential of the liquid blood concentrate Platelet-rich fibrin (PRF) as an adjunctive therapy for managing localized skin cancers, particularly NMSC” and Lines 71 – 72 “In this context, our study aims to investigate the potential of PRF as a possible adjunctive therapy” The authors here state as aim of their study the exploration of Use of PRF as adjunctive therapy, but their results do not provide evidence neither for “therapy” nor for “adjunctive therapy”. It is to note that the in vitro experiments do not provide any evidence about therapeutic effectiveness. Moreover, the exploration of “adjunctive therapies would need a very different study design. So, to be accurate what the authors may meant is to study the effects of “PRF in SCC and BCC cells in order to obtain results that may enhance the exploration of new therapies”. The aim must be rewritten.
Line 39 The phrase “collectively representing over 95% of cases” need citation or could be written as “collectively representing the vast majority of cases”
Lines 50 – 51 at the end of the sentence citation is missing
Line 55 Consider changing paragraph for “Platelet
Lines 57 – 65 This part fails to explain the rationale behind the experimentation. Why is PRF studied as an anticancer factor? (It is a core subject to explain why you did these experiments. Even if this study is preliminary, it is very interesting and the most interesting of all is the rationale behind using a factor without any cytotoxicity and having results of apoptosis of malignant cells)Maybe Transfer here lines 230 – 232?
Line 99 Diagrams D and E lack error bars. The Caption lacks reporting of number of replicates.
Line 121 The Caption lacks reporting of the number of replicates.
Figures 4 – 7 It is not understandable why the control bars lack error bars (only one specimen?) The caption also lacks reporting of the number of replicates
Lines 239 – 240 The claim is not supported especially from Figure 1 b and c diagrams
Lines 222 – 231 The discussion part contains information that would be better to be reported in the introduction. Moreover, A discussion part should first of all refer to the current study’s findings (discussing each figure and table) explaining the view of the authors why we have different viability patterns among the different cell types. For example, why cell viability results are relative to the results of proliferative active cells measurements and so on. In my view both Discussion and Conclusions part would benefit from a full rewrite.
Lines 326 – 327 Please state that the consent was written
Lines 329 – 334 Provide the catalog numbers of cell lines. Avoid abbreviations/acronyms without full explanation. Explain how many cell cultures per cell line you had.
Lines 339 – 341 Provide inclusion and exclusion criteria along with information about the donors (how many donors? Male Female? Mean and Median age? ecc).
Line 357 Report the number of replicates per experiment
Line 366 Report the microplate reader, along with producing company ecc
Line 379 “Eclipse Ni/E fluorescence microscope with a DS-Ri2 camera.” Report source of microscope and camera (company, ecc). Report the excitation and emission wavelengths and microscope filter settings
Lines 368 – 391 Report number of replicates. Also report what is the goal of this method (To detect proliferative active cells? Must be reported)
Line 393 Report manufacturer for RNeasy Micro Kit and any other device in the manuscript.
Lines 414 – 418 The part of statistical analysis cannot be assessed as the number of replicates measured in not reported in the materials and methods part. A general view is that to perform ANOVA you need first to perform Shapiro-wilk test for normality. Then Levene’s test for homogeneity of variances. If the results follow normal distribution and equity of variances, only then can ANOVA be considered as suitable. But if the replicates are up to 5 (that is very common practice) then the non-parametric tests would seem more suitable.
Author Response
Dear Reviewer,
we would like to thank you for carefully reading the manuscript and for your response. Please find included the resubmission of the manuscript with the revised title:
“Effect of liquid blood concentrates on cell proliferation and cell cycle- and apoptosis-related gene expressions in nonmelanoma skin cancer cells: A comparative in vitro study”
by Eva Dohle, Lianna Zhu, Robert Sader and Shahram Ghanaati,
to be considered for publication as original research paper in the International Journal of Molecular Science. We have revised the manuscript according to your suggestions. The changes are highlighted in yellow color in the revised version of the manuscript and addressed in this letter. We would like to thank you for all your effort with the manuscript.
Yours sincerely,
Eva Dohle
General information:
The individual answers to the reviewer’s suggestions are addressed in this letter point by point. All changes in the revised manuscript have been highlighted in yellow colour.
I went thorough the manuscript titled “Immunomodulating effect of liquid blood concentrates on cell proliferation and cell cycle- and apoptosis-related gene expressions in nonmelanoma skin cancer cells: A comparative in vitro study”. Although this experimental work relies on a highly innovative idea of using Platelet-rich fibrin on SCC and BCC as an anti-malignant factor, the manuscript presents several flaws and ambiguities that limit my ability to assess the quality of the work and the validity of the results.
Below, I describe my suggestions in detail.
Line 2 The Word “Immunomodulating” should be excluded. In this study the researchers used cell lines to see expressions and viability. Cell cultures lack immune system and are not a suitable model for immunomodulating effects of any kind. Consider changing the title to describe your work. Since you have used Platelet-rich fibrin on SCC and BCC cell lines and measured gene expressions and cell viability that is you should report to the title
According to this suggestion, the word ‘immunomodulating’ has been excluded from the title of the manuscript. The new title is as follows: “Effect of liquid blood concentrates on cell proliferation and cell cycle- and apoptosis-related gene expressions in nonmelanoma skin cancer cells: A comparative in vitro study”
Lines 15 – 17 “Our study aims to explore the potential of the liquid blood concentrate Platelet-rich fibrin (PRF) as an adjunctive therapy for managing localized skin cancers, particularly NMSC” and Lines 71 – 72 “In this context, our study aims to investigate the potential of PRF as a possible adjunctive therapy” The authors here state as aim of their study the exploration of Use of PRF as adjunctive therapy, but their results do not provide evidence neither for “therapy” nor for “adjunctive therapy”. It is to note that the in vitro experiments do not provide any evidence about therapeutic effectiveness. Moreover, the exploration of “adjunctive therapies would need a very different study design. So, to be accurate what the authors may meant is to study the effects of “PRF in SCC and BCC cells in order to obtain results that may enhance the exploration of new therapies”. The aim must be rewritten.
According to this suggestion, the authors decided to rewrite this paragraph and to generally formulate this paragraph more cautiously with regard to a possible adjunctive therapy. The aim of this study has been rewritten and described as “may enhance the exploration of new therapies”, respectively. The changes in lines 15-17 as well as in lines 71-72 have been highlighted in yellow color in the revised manuscript.
Line 39 The phrase “collectively representing over 95% of cases” need citation or could be written as “collectively representing the vast majority of cases”
The authors changed the phrase accordingly.
Lines 50 – 51 at the end of the sentence citation is missing
The appropriate reference has been added to the manuscript, accordingly.
Line 55 Consider changing paragraph for “Platelet
According to this, the authors decided to slightly shorten this paragraph.
Lines 57 – 65 This part fails to explain the rationale behind the experimentation. Why is PRF studied as an anticancer factor? (It is a core subject to explain why you did these experiments. Even if this study is preliminary, it is very interesting and the most interesting of all is the rationale behind using a factor without any cytotoxicity and having results of apoptosis of malignant cells)Maybe Transfer here lines 230 – 232?
According to the reviewer’s suggestion, the authors provided more information on why PRF was studied as an anticancer treatment. The revised paragraph has been highlighted in yellow color in the revised manuscript and has been additionally copied to this letter below:
„The bioactivity of PRF seems to be driven by its autogenous nature and the concentrated delivery of immune cells including platelets, leukocytes, and various plasma proteins embedded in a fibrin matrix are serving as a reservoir for growth factors and cytokines enabling to enhance immune responses. The understanding that tumor cells are cells that can evade immunity through mutations leads to the idea that a strengthened immune system combined with other treatments might help to eliminate the abnormal cells or prevent them from further growing “
Line 99 Diagrams D and E lack error bars. The Caption lacks reporting of number of replicates.
We added error bars with the SD to figure 1D and E as suggested. Furthermore, for better visualization of the differences in cell viability of PRF treated cells compared to untreated controls, we decided to add the control group (=100 %) to this graph as well. The number of replicates for this experiment have been also added to the figure legends according to the reviewer’s suggestion.
Line 121 The Caption lacks reporting of the number of replicates.
The number of replicates has been added to the figure legends of figure 2.
Figures 4 – 7 It is not understandable why the control bars lack error bars (only one specimen?) The caption also lacks reporting of the number of replicates
The results for relative gene expression are presented as fold change relative to RPL37A gene expression and relative gene expression was calculated using the delta delta CT method (by setting controls to 1 as reference value (RQ control =1)). We added a sentence with this information to the materials and methods section of the revised manuscript. Since data are calculated as relative to control, standard deviation cannot be shown for the control group. According to the reviewer’s suggestion, we added information on number of replicates in each figure legend.
Lines 239 – 240 The claim is not supported especially from Figure 1 b and c diagrams
The sentence has been rewritten accordingly.
Lines 222 – 231 The discussion part contains information that would be better to be reported in the introduction. Moreover, A discussion part should first of all refer to the current study’s findings (discussing each figure and table) explaining the view of the authors why we have different viability patterns among the different cell types. For example, why cell viability results are relative to the results of proliferative active cells measurements and so on. In my view both Discussion and Conclusions part would benefit from a full rewrite.
The discussion part has been shortened with regard to information that is better written in the introduction part as suggested. Furthermore, some additional statements/sentences have been written according to the reviewer’s suggestion.
Lines 326 – 327 Please state that the consent was written
This information has been added.
Lines 329 – 334 Provide the catalog numbers of cell lines. Avoid abbreviations/acronyms without full explanation. Explain how many cell cultures per cell line you had.
Accordingly, the catalog numbers and appropriate information on the cell lines has been added. In addition, information on number of cell cultures per cell line has been added as well.
Lines 339 – 341 Provide inclusion and exclusion criteria along with information about the donors (how many donors? Male Female? Mean and Median age? ecc).
The suggested information has been added to the revised version of the manuscript.
Line 357 Report the number of replicates per experiment
Number of replicates have been added.
Line 366 Report the microplate reader, along with producing company ecc
Has been added.
Line 379 “Eclipse Ni/E fluorescence microscope with a DS-Ri2 camera.” Report source of microscope and camera (company, ecc). Report the excitation and emission wavelengths and microscope filter settings
Information on fluorescence microscope, source of microscope, excitation and wavelength is included in the revised manuscript.
Lines 368 – 391 Report number of replicates. Also report what is the goal of this method (To detect proliferative active cells? Must be reported)
Number of replicates have been added accordingly. In addition, information regarding the goal of the method has been written in this section in the revised manuscript.
Line 393 Report manufacturer for RNeasy Micro Kit and any other device in the manuscript.
The appropriate manufacturers have been added to the revised version of the manuscript.
Lines 414 – 418 The part of statistical analysis cannot be assessed as the number of replicates measured in not reported in the materials and methods part. A general view is that to perform ANOVA you need first to perform Shapiro-wilk test for normality. Then Levene’s test for homogeneity of variances. If the results follow normal distribution and equity of variances, only then can ANOVA be considered as suitable. But if the replicates are up to 5 (that is very common practice) then the non-parametric tests would seem more suitable.
For each group, samples from three independent donors (n=3 PRF donors) were analyzed with three independent experiments conducted per donor (n=3 cell cultures per group and donor, means in total n=9). The data were collected (as part of a Real-Time PCR, MTS and KI67 staining experiments) across three groups (high PRF, low PRF, control), with the control group normalized to a value of 1 or 100. As the Shapiro-Wilk test indicated that the data in at least one group were not normally distributed (p < 0.05), the use of a classical Two-Way ANOVA is not appropriate. This method assumes normal distribution of the data within each group and homogeneity of variances between groups. Instead, a non-parametric test was employed, as it is more robust against deviations from the assumption of normality. Therefore, we used the One-Way ANOVA followed by post-hoc analyses (multiple comparison) during this study. According to the reviewer’s suggestion, we provided information regarding the number of donors and replicates and the statistical methods in the revised version of the manuscript.

Reviewer 2 Report (New Reviewer)
Comments and Suggestions for Authors
i read with great interest the manuscript regarding PRF in NMSC patients- at first very nice and interesting topic.
some comments:
ABstract: Report what low RCF is..
also report briefly your methods section- you present the results directly confusing the reader
Intro: more implications of PRF in dermatology and thhe pathophysiology impliactaions should be assessed
can PRF from patients with NMSC be an ideal specimen?? what can be special considerations as blood of NMSC patients have increased oxidative stress(doi.org/10.3390/stresses3040054) and systemic inflammation(doi.org/10.5114/aoms/177345) due to the systemic impact of cumulative sun exposure and NMSC cancer cells by products entering bloodstream.
methods and results
give a terminology of high and low RCF
I want more on Ki67 utility and marker significance
in Gigure 7 add the p53 in the diagrams
generally nice and organised schemes - i suggest a summary table of BCC and SCC observation in each case providing a more clear presenation of your findings
and also a figure with cells and arrows showing what happens in each case would be also a nice addition
in the discussion section add the clinical implications of your findings and write more the limiattions (not just the the in vitro nature)
generally nice manuscript
Author Response
Dear Reviewer,
we would like to thank you for carefully reading the manuscript and for your response. Please find included the resubmission of the manuscript with a revised title:
“Effect of liquid blood concentrates on cell proliferation and cell cycle- and apoptosis-related gene expressions in nonmelanoma skin cancer cells: A comparative in vitro study”
by Eva Dohle, Lianna Zhu, Robert Sader and Shahram Ghanaati,
to be considered for publication as original research paper in the International Journal of Molecular Science. We have revised the manuscript according to your suggestions. The changes are highlighted in yellow color in the revised version of the manuscript and addressed in this letter. We would like to thank you for all your effort with the manuscript.
Yours sincerely,
Eva Dohle
General information:
The individual answers to the reviewers’ suggestions are addressed in this letter point by point. All changes in the revised manuscript have been highlighted in yellow colour.
i read with great interest the manuscript regarding PRF in NMSC patients- at first very nice and interesting topic.
Thank you very much!
some comments:
ABstract: Report what low RCF is..
also report briefly your methods section- you present the results directly confusing the reader
According to these suggestions, the authors defined and reported the term ‘low PRF’ and also modified the abstract with regard to the used methods.
Intro: more implications of PRF in dermatology and thhe pathophysiology impliactaions should be assessed
The authors exemplarily added more information on PRF applications in dermatology according to this suggestion.
can PRF from patients with NMSC be an ideal specimen?? what can be special considerations as blood of NMSC patients have increased oxidative stress(doi.org/10.3390/stresses3040054) and systemic inflammation(doi.org/10.5114/aoms/177345) due to the systemic impact of cumulative sun exposure and NMSC cancer cells by products entering bloodstream.
This is a very interesting point/question. We decided to add a brief paragraph with regard to this topic and possible limitations of the study to the discussion part of the revised manuscript.
methods and results
give a terminology of high and low RCF
According to this suggestion, the authors provided the terminology and preparation protocols for low and high RCF PRF within the revised version of the materials and methods section.
I want more on Ki67 utility and marker significance
According to this suggestion, information regarding the goal of the method has been written in this section in the revised manuscript
in Gigure 7 add the p53 in the diagrams
The genes that were analysed in the appropriate diagrams are described and labeled in the appropriate figure legend of each figure.
generally nice and organised schemes - i suggest a summary table of BCC and SCC observation in each case providing a more clear presenation of your findings
and also a figure with cells and arrows showing what happens in each case would be also a nice addition
We agree that this might improve the understanding of the results. Nevertheless, since we have already reached the limit of permitted illustrations per manuscript, we have decided not to publish any additional illustrations after careful consideration, respectively. However, we will contact the editor to clarify whether another illustration might be possible and might improve the manuscript (possibly presented as a graphical abstract?).
in the discussion section add the clinical implications of your findings and write more the limiattions (not just the the in vitro nature)
We added a paragraph concerning the clinical limitations of our findings according to this suggestion.
generally nice manuscript
Thank you!
Round 2
Reviewer 1 Report (New Reviewer)
Comments and Suggestions for Authors
Dear Authors
Thank you for adressing my concerns
Reviewer 2 Report (New Reviewer)
Comments and Suggestions for Authors
The authors did take my suggestions into account and the manuscript is improved!! Well done
This manuscript is a resubmission of an earlier submission. The following is a list of the peer review reports and author responses from that submission.
Round 1
Reviewer 1 Report
Comments and Suggestions for Authors
The article by Eva Dohle and colleagues provides a comparative in vitro study on the immunomodulating effect of liquid blood concentrates on cell proliferation and cell cycle and apoptosis-related gene expressions in nonmelanoma skin cancer cells. Nonmelanoma skin cancer (NMSC) constitutes the following predominant subtypes: Basal cell carcinoma (BCC) and squamous cell carcinoma (SCC), which were characterized mainly by a high burden of mutations induced by ultraviolet radiation. The current treatment for NMSC has limitations, such as invasiveness and side effects, including scarring, pain, and local skin reactions, leading to novel treatments to address these challenges. The study's primary purpose was to examine PRF's potential as an adjunctive therapy for treating NMSC. In this study, the authors used multiple experiments to treat BCC and SCC cells with platelet-rich fibrin and study its effects on NMSC cells. The authors have performed different assays to investigate the study aim, such as cell viability, immunofluorescence staining with image quantifications, gene expression analyses, etc.
Overall, this is a well-written article, and the study's results highlight the antiproliferative effect of PRF in BCC and SCC cells by inducing cell cycle rest at the early treatment stages. The results further showed that in BSS and SCC, the downregulation of the negative cell cycle regulator CCND1, while in SCC cells, only the upregulation of the tumor suppressor genes p53 and p21. However, the following minor points will help improve the study findings.
The reference list is comprehensive and includes relevant publications. The authors must include the most recent publications (2023-2024) in this article and discuss the most recent findings here.
The authors should also mention non-significance appropriately in the statistical analysis section and all the figures in this article.
In Figure 2, the scale bar is not visible in the main figure or a region of interest. It would be a good idea if the authors mentioned the scale bar in the figure description. Also, in Figure 2B, some cell's nuclei appear more prominent than the rest of the cells. It would be great if the authors could comment on this. The authors mention that the cell detection command in QuPath was used to identify the number of cells in all nuclei based on nuclear staining in the blue DAPI channel in the materials and methods section. How does this affect the intensity threshold of detected cells with bigger size nuclei?
Author Response
Dear Reviewer,
we would like to thank you for carefully reading the manuscript and for your response. Please find included the resubmission of the manuscript entitled:
“Immunomodulating effect of liquid blood concentrates on cell proliferation and cell cycle- and apoptosis-related gene expressions in nonmelanoma skin cancer cells: A comparative in vitro study”
by Eva Dohle, Lianna Zhu, Robert Sader and Shahram Ghanaati,
to be considered for publication as original research paper in the International Journal of Molecular Science. We have revised the manuscript according to your suggestions. The changes are addressed in this letter. We would like to thank you for all your effort with the manuscript.
Yours sincerely,
Eva Dohle
General information:
The individual answers to the reviewers’ suggestions are addressed in this letter. All changes in the revised manuscript have been highlighted in yellow colour.
The article by Eva Dohle and colleagues provides a comparative in vitro study on the immunomodulating effect of liquid blood concentrates on cell proliferation and cell cycle and apoptosis-related gene expressions in nonmelanoma skin cancer cells. Nonmelanoma skin cancer (NMSC) constitutes the following predominant subtypes: Basal cell carcinoma (BCC) and squamous cell carcinoma (SCC), which were characterized mainly by a high burden of mutations induced by ultraviolet radiation. The current treatment for NMSC has limitations, such as invasiveness and side effects, including scarring, pain, and local skin reactions, leading to novel treatments to address these challenges. The study's primary purpose was to examine PRF's potential as an adjunctive therapy for treating NMSC. In this study, the authors used multiple experiments to treat BCC and SCC cells with platelet-rich fibrin and study its effects on NMSC cells. The authors have performed different assays to investigate the study aim, such as cell viability, immunofluorescence staining with image quantifications, gene expression analyses, etc.
Overall, this is a well-written article, and the study's results highlight the antiproliferative effect of PRF in BCC and SCC cells by inducing cell cycle rest at the early treatment stages. The results further showed that in BSS and SCC, the downregulation of the negative cell cycle regulator CCND1, while in SCC cells, only the upregulation of the tumor suppressor genes p53 and p21. However, the following minor points will help improve the study findings.
The reference list is comprehensive and includes relevant publications. The authors must include the most recent publications (2023-2024) in this article and discuss the most recent findings here.
We would like to thank the reviewer for this suggestion. After a new and extensive literature research (Pubmed), we unfortunately found no current publications (2023-2024) on the subject of ‘effect of blood concentrates on cancer cells’ apart from our own publication from this year (2024; Dohle et al.). This publication has been already cited in various places in the manuscript.
The authors should also mention non-significance appropriately in the statistical analysis section and all the figures in this article.
According to this suggestion, the authors revised the results section and added more and additional information about the non-significance of the different results in the appropriate sections. The changes are marked in yellow colour.
In Figure 2, the scale bar is not visible in the main figure or a region of interest. It would be a good idea if the authors mentioned the scale bar in the figure description.
According to this suggestion, we mentioned the scale bar with the appropriate sizes in the figure legend (marked in yellow colour).
Also, in Figure 2B, some cell's nuclei appear more prominent than the rest of the cells. It would be great if the authors could comment on this. The authors mention that the cell detection command in QuPath was used to identify the number of cells in all nuclei based on nuclear staining in the blue DAPI channel in the materials and methods section. How does this affect the intensity threshold of detected cells with bigger size nuclei?
Thank you for this suggestion/question. The authors performed nuclei staining to detect or to identify the number of cells in total. The software calculated the number of cell nuclei in each randomized chosen figure automatically according to the intensity, independent of the size of the nucleus. Cells were classified as Ki67 positive if the intensity threshold of detected cells was above 100 in the red Ki67 channel of the image. The percentage of positive cells calculated from the PRF treated group was compared to that of the control group for analysis. Since the cell viability assay determined less viable cells in response to PRF treatment, solely Ki67 staining (without referring to the total amount of cell nuclei) would not accurately reflect the results in terms of differences in Ki67 expression between the different experimental groups. We revised this part of the methods section and added more information on the quantification method using QuPath software (marked in yellow colour).

Reviewer 2 Report
Comments and Suggestions for Authors
The manuscript entitled “Immunomodulating effect of liquid blood concentrates on cell proliferation and cell cycle- and apoptosis-related gene expressions in nonmelanoma skin cancer cells: A comparative in vitro study” by Eva Dohle et al., was presented as an Original Article belonging to the “Molecular Pathology, Diagnostics, and Therapeutics” section (Special Issue: A Commemorative Special Issue in Honor of Prof. Giovanni De Toni: A Pediatrician and Innovator at the Gaslini Children’s Hospital, Genoa, Italy). By using in vitro BCC and SCC cell lines, the authors aimed to provide evidence about the potential use of Platelet Rich Fibrin (PRF) as an adjunctive therapy for treatment of non-melanoma skin cancers, taking advantage its anti-proliferative effects. Although the thematic of the research is relevant and the article could be of potential interest not only for a wide community of basic researchers but also for clinicians, in the current state the manuscript appears still preliminary, and the quality of the data does not reach the expected quality standards required for publication. Several major points need to be addressed to strengthen the technical rigor.
Comments for the data:
In general, the data presented in the figures need to be improved. One main limitation of the study is that the validation methods are not abundant enough. Therefore, to support the conclusions, the results should be improved including several experimental approaches and different methodologies.
1. Using just one cell line as model of BCC (BCC-1/KMC) and SCC (SCC-25) in the study is inadequate. It's recommended to incorporate a minimum of two different cell lines for more comprehensive results, especially because it is a study exclusively in vitro.
2. The gene expression analysis of Cyclin-D1, p21, p53, Caspase 8 and Caspase 9 was normalized against a single housekeeping gene, which is somewhat unusual. Have the authors tested other housekeeping genes to confirm stability of RPL37A (Ribosomal Protein L37a) under the treatment conditions?
3. Although the authors reported that treatment with low and high dose of PRF impairs the expression of Cyclin-D1, p21 and p53, the gene expression analyses should be confirmed by the protein expression of the same genes.
Comments for the figures:
1. The size of immunofluorescence analyses is too small. Cellular morphology and characteristics are not evident in the current format. Please, resize and provide a zoom of a representative field. Information about magnification (x) is missing; add and specify it in the Figure Legend.
Author Response
Dear Reviewer,
we would like to thank you for carefully reading the manuscript and for your response. Please find included the resubmission of the manuscript entitled:
“Immunomodulating effect of liquid blood concentrates on cell proliferation and cell cycle- and apoptosis-related gene expressions in nonmelanoma skin cancer cells: A comparative in vitro study”
by Eva Dohle, Lianna Zhu, Robert Sader and Shahram Ghanaati,
to be considered for publication as original research paper in the International Journal of Molecular Science. We have revised the manuscript according to your suggestions. The changes are addressed in this letter. We would like to thank you for all your effort with the manuscript.
Yours sincerely,
Eva Dohle
General information:
The individual answers to the reviewers’ suggestions are addressed in this letter. All changes in the revised manuscript have been highlighted in yellow colour.
The manuscript entitled “Immunomodulating effect of liquid blood concentrates on cell proliferation and cell cycle- and apoptosis-related gene expressions in nonmelanoma skin cancer cells: A comparative in vitro study” by Eva Dohle et al., was presented as an Original Article belonging to the “Molecular Pathology, Diagnostics, and Therapeutics” section (Special Issue: A Commemorative Special Issue in Honor of Prof. Giovanni De Toni: A Pediatrician and Innovator at the Gaslini Children’s Hospital, Genoa, Italy). By using in vitro BCC and SCC cell lines, the authors aimed to provide evidence about the potential use of Platelet Rich Fibrin (PRF) as an adjunctive therapy for treatment of non-melanoma skin cancers, taking advantage its anti-proliferative effects. Although the thematic of the research is relevant and the article could be of potential interest not only for a wide community of basic researchers but also for clinicians, in the current state the manuscript appears still preliminary, and the quality of the data does not reach the expected quality standards required for publication. Several major points need to be addressed to strengthen the technical rigor.
Comments for the data:
In general, the data presented in the figures need to be improved. One main limitation of the study is that the validation methods are not abundant enough. Therefore, to support the conclusions, the results should be improved including several experimental approaches and different methodologies.
The results of this study have been very carefully evaluated and show, using various methods, that PRF seems to have a suppressive effect on cell growth and tumor cell turnover. These results were in accordance with a former study of our group (Dohle et al 2024). As written in the manuscript, the study is a basic study and the results have been very carefully and cautiously placed in the overall context. It is a great pity that this was apparently not recognized. Accordingly, some passages of the manuscript have been revised and adapted again. Changes are highlighted in yellow colour in the revised manuscript.
- Using just one cell line as model of BCC (BCC-1/KMC) and SCC (SCC-25) in the study is inadequate. It's recommended to incorporate a minimum of two different cell lines for more comprehensive results, especially because it is a study exclusively in vitro.
The authors wanted to analyse the effect of PRF on two different NMSC cell types, namely BCC and SCC. The most commonly used cell types for studying non-melanoma skin cancers are derived from basal cell carcinoma (BCC) and squamous cell carcinoma (SCC), which are the two main types of non-melanoma skin cancer. The cell lines serve as models for research and are the most commonly used cell types in the scientific community with regard to NMSC research and a tool for understanding the biology of non-melanoma skin cancers and for testing potential therapeutic agents. Although the authors decided not to additionally analyse the PRF mediated effect on another cell variant of NMSC, we’ll keep this in mind for a following study and thank the reviewer for this idea.
- The gene expression analysis of Cyclin-D1, p21, p53, Caspase 8 and Caspase 9 was normalized against a single housekeeping gene, which is somewhat unusual. Have the authors tested other housekeeping genes to confirm stability of RPL37A (Ribosomal Protein L37a) under the treatment conditions?
During this study, we decided to normalize the gene expression of the genes of interest (RQ) relative to the housekeeping gene RPL13A because we assessed this housekeeping gene as stable under treatment conditions. In the initial experiments, we first analysed gene expression of different housekeeping genes and could estimate RPL13A as the most stable gene in response to PRF treatment.
- Although the authors reported that treatment with low and high dose of PRF impairs the expression of Cyclin-D1, p21 and p53, the gene expression analyses should be confirmed by the protein expression of the same genes.
We would like to thank the reviewer for this suggestion. To our experience and especially for our study purposes, analyzing gene expression rather than protein expression has certain advantages for our research objectives. Key reasons why we did choose gene expression analysis instead of protein expression analyses in this study are the easier and faster quantification process and the higher sensitivity. Furthermore, gene expression analysis provides a broader view of transcriptional regulation. Protein expression, on the other hand, is more specific to the actual functional molecules, but many regulatory events occur at the RNA level, which protein analysis may not detect. In addition, protein expression requires more complex methods like Western blotting which are often more intensive, less sensitive, and may not be as high-throughput.
Comments for the figures:
- The size of immunofluorescence analyses is too small. Cellular morphology and characteristics are not evident in the current format. Please, resize and provide a zoom of a representative field. Information about magnification (x) is missing; add and specify it in the Figure Legend.
Thank you for this suggestion/question. The immunofluorescence staining pictures in figure 2 are representatives for the quantification of Ki67 positive cells (relative to the total cell nuclei amount of the different experimental groups; figure 2C and D). In this regard, the dapi positive cells as well as the Ki67 positive cells can be very well recognized and are well visible; especially in the zoom pictures in the lower rows. The authors performed nuclei staining to detect or to identify the number of cells in total. The software calculated the number of cell nuclei in each randomized chosen figure automatically. Cells were classified as Ki67 positive if the intensity threshold of detected cells was above 100 in the red Ki67 channel of the image. The percentage of positive cells calculated from the PRF treated group was compared to that of the control group for analysis. We revised this part of the methods section and added more information on the quantification method using QuPath software (marked in yellow colour). In addition and according to the reviewers’ suggestion, we mentioned the scale bar with the appropriate sizes in the figure legend (marked in yellow colour).

Reviewer 3 Report
Comments and Suggestions for Authors
Dohle and colleagues presented a research article aimed at assessing the therapeutic potential of liquid blood concentrates on non-melanoma skin cancer. For this purpose, the authors performed some preliminary in vitro analyses demonstrating the role of blood concentrates on cell proliferation and cell cycle- and apoptosis-related gene expressions. Although interesting, the manuscript needs major revisions to be accepted for publication. Please see and carefully address the comments reported below:
1) It is not clear how the number of Ki67 positive cells were calculated. Please give more details in the methods or results sections;
2) Do not use separate subsections to describe the results obtained for caspase 8, caspase 9, cyclin D1, etc.;
3) How do you explain the fluctuations of CCND1 gene expression observed in Figure 5? Specifically, CCND1 levels are low at first, then increase and decrease again in SCC cells (2nd, 4th and 7th day, respectively). The same conflicting data were obtained for BCC at 4 and 7 days. This is very confounding;
4) Please check the errors in Figure 5 panel F. On the top you should indicate “7d” while on the bottom you should put “SCC”. Please check and correct;
5) Figure 7 appears with a different style and layout compared to the other Figures. Please use the same border and line thickness;
In the Discussion section, the authors provided a broad description of the interactions between PRF and immune cells. However, none of these interactions were investigated in the manuscript. My suggestion is to significantly shorten this part and focus on the data obtained in the present manuscript;
6) Please better discuss the results presented from line 299 to line 328. You just reported the obtained results on gene expression. However, you should better describe the different mutations and signaling pathway alterations observed in NMSC providing appropriate references. For this purpose, please see:
- PMID: 32899768
- PMID: 35625975
- PMID: 35187675
7) Please provide the catalog numbers of the antibodies used for immunofluorescence experiments;
8) Please, move the information provided in section 4.7 within section 4.6 as these are the same experiments;
9) In the Methods section, provide a Table or Supplementary Table containing the sequences of the primers used for RT-qPCR experiments;
10) A key limitation of the study is the lack of confirmatory experiments on the protein expression of the genes tested through RT-qPCR. Did you have any western blot data? Please give a comment on this.
Author Response
Dear Reviewer,
we would like to thank you for carefully reading the manuscript and for your response. Please find included the resubmission of the manuscript entitled:
“Immunomodulating effect of liquid blood concentrates on cell proliferation and cell cycle- and apoptosis-related gene expressions in nonmelanoma skin cancer cells: A comparative in vitro study”
by Eva Dohle, Lianna Zhu, Robert Sader and Shahram Ghanaati,
to be considered for publication as original research paper in the International Journal of Molecular Science. We have revised the manuscript according to your suggestions. The changes are addressed in this letter. We would like to thank you for all your effort with the manuscript.
Yours sincerely,
Eva Dohle
General information:
The individual answers to the reviewers’ suggestions are addressed in this letter. All changes in the revised manuscript have been highlighted in yellow colour.
Dohle and colleagues presented a research article aimed at assessing the therapeutic potential of liquid blood concentrates on non-melanoma skin cancer. For this purpose, the authors performed some preliminary in vitro analyses demonstrating the role of blood concentrates on cell proliferation and cell cycle- and apoptosis-related gene expressions. Although interesting, the manuscript needs major revisions to be accepted for publication. Please see and carefully address the comments reported below:
- It is not clear how the number of Ki67 positive cells were calculated. Please give more details in the methods or results sections;
According to this suggestion, more details concerning the method to calculate the number of Ki67 positive cells have been added in the revised version of the manuscript. Changes have been addressed in yellow colour.
- Do not use separate subsections to describe the results obtained for caspase 8, caspase 9, cyclin D1, etc.;
According to this suggestion, the subsection (subheadings) for the gene expression analyses have been removed in the revised manuscript.
- How do you explain the fluctuations of CCND1 gene expression observed in Figure 5? Specifically, CCND1 levels are low at first, then increase and decrease again in SCC cells (2nd, 4th and 7th day, respectively). The same conflicting data were obtained for BCC at 4 and 7 days. This is very confounding;
In general, the combination of PRF with its cellular components and the PRF-mediated effect on NMSCs is highly regulated and complex since the different cell types strongly act together as a unique system in which each cell type is influencing the other one through paracrine or direct communication including complicated feedback mechanisms. The authors analysed gene expression after different time points (2d, 4d, 7d). It is well known that gene expression changes over time during treatment might be due to several dynamic biological processes. The time-dependent alterations in gene expression might reflect the cell's ability to adapt, resist, or recover from the effects of the treatment. In addition, cells in different phases of the cell cycle may have different levels of transcriptional and translational machinery, affecting gene expression. Although the authors did not analyze the molecular reasons/background for the fluctuations in gene expression over the treatment time, the changes might be also dose-dependent, especially with regard to the changes in gene expression from 2d to 4d. The fluctuations between gene expression after 4d compared to 7d might then be explained with the activation of secondary responses. Nevertheless, a follow-up study will be planed to further evaluate the molecular background and underlying mechanism of PRF-mediated effects on NMSCs.
- Please check the errors in Figure 5 panel F. On the top you should indicate “7d” while on the bottom you should put “SCC”. Please check and correct;
According to this we checked and corrected the incorrect labeling of figure 5F.
- Figure 7 appears with a different style and layout compared to the other Figures. Please use the same border and line thickness;
The style of figure 7 has been adjusted and corrected accordingly and according to the other figures.
- In the Discussion section, the authors provided a broad description of the interactions between PRF and immune cells. However, none of these interactions were investigated in the manuscript. My suggestion is to significantly shorten this part and focus on the data obtained in the present manuscript; Please better discuss the results presented from line 299 to line 328. You just reported the obtained results on gene expression. However, you should better describe the different mutations and signaling pathway alterations observed in NMSC providing appropriate references.
According to this suggestion, the discussion part describing the effect of immune cells as cellular component of PRF has been reduced and shortened. In addition, the authors added some information on the signaling pathway alterations in NMSC with the appropriate references. Changes are marked in yellow colour in the revised manuscript.
7) Please provide the catalog numbers of the antibodies used for immunofluorescence experiments;
The company as well as the catalog number has been added to the revised manuscript.
8) Please, move the information provided in section 4.7 within section 4.6 as these are the same experiments;
The sections 4.6 and 4.7 have been summarized to one section 4.6 according to the reviewer’s suggestion.
9) In the Methods section, provide a Table or Supplementary Table containing the sequences of the primers used for RT-qPCR experiments;
A table containing the sequences of the primers and the catalog numbers of the used primer assays has been added to the method section according to this suggestion.
10) A key limitation of the study is the lack of confirmatory experiments on the protein expression of the genes tested through RT-qPCR. Did you have any western blot data? Please give a comment on this.
We would like to thank the reviewer for this suggestion. To our experience and especially for our study purposes, analyzing gene expression rather than protein expression has certain advantages for our research objectives. Key reasons why we did choose gene expression analysis instead of protein expression analyses in this study are the easier and faster quantification process and the higher sensitivity. Furthermore, gene expression analysis provides a broader view of transcriptional regulation. Protein expression, on the other hand, is more specific to the actual functional molecules, but many regulatory events occur at the RNA level, which protein analysis may not detect. In addition, protein expression requires more complex methods like Western blotting which are often more intensive, less sensitive, and may not be as high-throughput.

Round 2
Reviewer 2 Report
Comments and Suggestions for Authors
The reviewer is surprised that the authors did not even try to respond to the reviewer’s requests.
With regard to previous reviewer’s COMMENT #1: Using just one cell line as model of BCC (BCC-1/KMC) and SCC (SCC-25) in the study is inadequate. It's recommended to incorporate a minimum of two different cell lines for more comprehensive results, especially because it is a study exclusively in vitro the authors have arbitrarily chosen to not additionally analyze the PRF mediated effect on almost a second cell variant for both BCC and SCC. Just because the cell lines serve as models for research and represent only a tool for understanding the biology of cancer in general and for testing potential therapeutic agents, considering the wide mutational spectrum between cell lines (even if they belong to the same type of cancer) it is unacceptable support the article conclusions. It would have been different if the authors had presented in vivo experiments.
With regard to previous reviewer’s COMMENT #2: The gene expression analysis of Cyclin-D1, p21, p53, Caspase 8 and Caspase 9 was normalized against a single housekeeping gene, which is somewhat unusual. Have the authors tested other housekeeping genes to confirm stability of RPL37A (Ribosomal Protein L37a) under the treatment conditions? the authors answered that during the study they analyzed the gene expression of different housekeeping genes and estimated RPL13A as the most stable gene in response to PRF treatment. However, probably they misunderstood the requirement. It is necessary to use almost two different housekeeping genes to confirm the results of qRT-PCR. Thus, in addition to the RPL37A normalization, the authors should normalize the expression of target genes against another housekeeping gene (for example alpha Tubulin, beta Actin, ribosome small subunit 18S, Cyclophin-A or similar) and show the results also as Supplemental Data.
With regard to previous reviewer’s COMMENT #3: Although the authors reported that treatment with low and high dose of PRF impairs the expression of Cyclin-D1, p21 and p53, the gene expression analyses should be confirmed by the protein expression of the same genes the authors answered that, based on their own experience, gene expression analyses are better than protein expression analyses. The reviewer supports the choice of authors, but just because gene expression analysis provides a broader view of transcriptional regulation networks (that in the present study are missing), it would be helpful perform an RNA-seq analysis in order to clarify the PRF-mediated Gene Ontology alterations.
With regard to previous reviewer’s COMMENT #4: The size of immunofluorescence analyses is too small. Cellular morphology and characteristics are not evident in the current format. Please, resize and provide a zoom of a representative field. Information about magnification (x) is missing; add and specify it in the Figure Legend the authors refused to show a representative zoom of the single panels, believing that the images shown were sufficient. To better show the Ki-67 signal, it would be better a single cell magnification (63X).
In conclusion, the reviewer does not approve the manuscript for the publication in the current state since a major revision is required for the reaching of the expected quality standards necessary for a publication. The editor has the final say on whether or not to accept the manuscript in its present condition.